# The Implementation of National Action Plan (NAP) on Antimicrobial Resistance (AMR) in Bangladesh: Challenges and Lessons Learned from a Cross-Sectional Qualitative Study

**DOI:** 10.3390/antibiotics11050690

**Published:** 2022-05-19

**Authors:** Syed Masud Ahmed, Nahitun Naher, Samiun Nazrin Bente Kamal Tune, Bushra Zarin Islam

**Affiliations:** Centre of Excellence for Health Systems and Universal Health Coverage, BRAC James P Grant, School of Public Health (JPGSPH), BRAC University, Dhaka 1213, Bangladesh; nahitun.naher@bracu.ac.bd (N.N.); samiun.tune@bracu.ac.bd (S.N.B.K.T.); bushra.zarin@bracu.ac.bd (B.Z.I.)

**Keywords:** National Action Plan on antimicrobial resistance, human and animal health, environment, COVID-19, Bangladesh

## Abstract

This study explored the current situation of the National Action Plan (NAP) on Antimicrobial Resistance (AMR) implementation in Bangladesh and examined how different sectors (human, animal, and environment) addressed the AMR problem in policy and practice, as well as associated challenges and barriers to identifying policy lessons and practices. Informed by a rapid review of the available literature and following the World Health Organization (WHO) AMR situation analysis framework, a guideline was developed to conduct in-depth interviews with selected stakeholders from January to December 2021. Data were analysed using an adapted version of Anderson’s governance framework. Findings reveal the absence of required inter-sectoral coordination essential to a multisectoral approach. There was substantial coordination between the human health and livestock/fisheries sectors, but the environment sector was conspicuously absent. The government initiated some hospital-based awareness programs and surveillance activities, yet no national Monitoring and Evaluation (M&E) framework was established for NAP activities. Progress of implementation was slow, constrained by the shortage of a trained health workforce and financial resources, as well as the COVID-19 pandemic. To summarise, five years into the development of the NAP in Bangladesh, its implementation is not up to the level that the urgency of the situation requires. The policy and practice need to be cognisant of this fact and do the needful things to avoid a catastrophe.

## 1. Introduction

Antimicrobial agents (including antibiotic, antiviral, antifungal and antiprotozoal) are critical tools for fighting diseases in humans, terrestrial and aquatic animals and plants. Alarming levels of resistance to these have been reported in countries of all income levels. As a result, common diseases are becoming difficult to treat, and lifesaving medical procedures are becoming riskier to perform [1,2]. Antimicrobial resistance (AMR) is increasingly becoming a global health and development threat, and the World Health Organisation (WHO) declared AMR as one of the top 10 global public health threats facing humanity [3]. This requires urgent multisectoral action to achieve the Sustainable Development Goals (SDGs) and universal health coverage (UHC).

Misuse and overuse of antimicrobials are the main drivers of AMR development [4]. These activities include irrational use of antimicrobials in humans and animals, improper storage and disposal of antimicrobials and inappropriate use of antibacterial agents in household products [5,6]. The number of novel antibiotics is diminishing, risking the rise of untreatable infections and the inevitable loss of life, especially in resource-constrained countries with limited treatment options [7]. This problem is compounded by poor access to necessary antimicrobials, poor surveillance of their use and resistance, a poor regulatory regime, lack of updated guidelines and continuing medical education for prescribers, and self-medication [8,9,10]. To address these challenges, the Global Action Plan (GAP) on AMR was launched at the World Health Assembly in 2015, focusing global attention on the need to design multisectoral national AMR action plans [11]. At present, several countries have initiated GAPs; however, the opportunity to investigate and inform the process for the engagement of policy bodies and national stakeholders remains limited [8,12].

Southeast Asia is considered to have the highest risk of AMR among all the WHO regions [13]. In many of these countries, such as Bangladesh, antimicrobials are widely available as over-the-counter (OTC) drugs from unregulated drug retail outlets (‘drug shops’) [14,15,16]. This situation is aggravated by the pharmaceutical companies’ aggressive and unethical marketing practices to boost sales, directed at the drug shop attendants and other informal providers [17]. Additionally, the regulatory regime in Bangladesh concerning human, technical and logistic capacity to oversee this vast market is weak [15,18].

Bangladesh has developed and approved its National Action Plan (NAP) for containing AMR from a One Health perspective and aligned it with the WHO GAP guidelines [19]. To date, hardly any studies have examined the contextual drivers of AMR policy development, assessed implementation status and consolidated lessons learned from the policy actors’ perspectives. This study was undertaken to explore the current situation of NAP on AMR implementation in Bangladesh and improve understanding of the dynamics of its development, constraints of implementation and perceived measures to overcome these in policy and practice.

### Conceptual Framework

The complex nature of AMR demands a comprehensive framework for assessing a range of barriers in implementing the NAP for antimicrobial use. A systematic content analysis of NAPs from the Association of Southeast Asian Nations (ASEAN) countries has used an adapted version of the Anderson framework [20]. We propose using this framework, developed as the first governance framework, to offer guidance for developing and assessing national action plans on AMR aligned with the GAP. Themes were broadly classified under five governance areas: (a) policy design (participation, policy development, coordination across multiple sectors, equity and transparency); (b) implementation tools (surveillance, stewardship and infection prevention and control measures); (c) monitoring and evaluation; (d) sustainability (resource allocation and availability of budget); and (e) One Health engagement (involvement of human, animal and environmental health sectors) for the implementation of NAP. Considering the dynamic nature of AMR, the framework is conceptualised as a cyclical process and is responsive to the context and allows for continuous improvement and adaptation of NAP on AMR (Figure 1).

## 2. Materials and Methods

We adopted a cross-sectional qualitative design to review the current state of the implementation of the NAP to contain AMR and elicit the perspectives of the policymakers and stakeholders on this issue. Informed by a rapid review of the available literature on the topic, we conducted in-depth interviews (IDIs) of selected stakeholders from the ministry, regulatory body and relevant institutions.

### 2.1. Sampling Frame and Sample

We applied a purposive sampling technique to identify and reach relevant respondents. Based on document review, suggestions by IDI guidelines, and experience with AMR networks from past studies, we prepared a tentative list to map and identify relevant organisations and policy stakeholders. Respondents were purposively selected from various public and private organisations in the human, animal, fisheries and environment sectors. For example, the Communicable Disease Control (CDC) of the Directorate General of Health Services (DGHS), the Ministry of Health and Family Welfare (MoHFW); the Department of Livestock Services (DLS), the Department of Environment (DoE), One Health Coordinator, Government of Bangladesh (GoB); the United States Agency for International Development (USAID) supported medicines, technologies and pharmaceutical services programs, epidemiological and livestock research institutions, food and agriculture organisations and medical college hospitals etc.

### 2.2. Tool Development

We developed the IDI guideline following the Anderson framework [20] and WHO AMR Situation Analysis Framework [14] for assessing NAP progress. We began by recording the respondents’ socio-demographic information. The main section included nine domains incorporating the above frameworks’ core components. Besides this, three additional domains explored respondents’ AMR work experience, the impact of COVID-19 on NAP implementation and a brief stakeholder analysis. Thus, the IDI guideline had 12 domains in total, as shown in Table 1.

### 2.3. Ethical Approval

This study received ethical approval from the Institutional Review Board of the BRAC James P Grant School of Public Health, BRAC University (Reference No. IRB-23 February’21-003). Informed voluntary verbal consent, including consent to record the conversation, was obtained from the respondents before the interviews. Participants were informed about the purpose, process, risks and benefits of the study. Respondents were given a choice to withdraw from the interview if they wanted. The respondents’ privacy, anonymity and confidentiality were maintained throughout the study, and data were used for research purposes only

### 2.4. Data Collection and Management

We approached the listed policy actors through an email invitation to participate in the study. We shared information on the project’s background, aims and objectives in the email invitation. Follow-up phone calls confirmed and fixed the appointment for the interview. We conducted 17 IDIs in person or through telephone calls based on respondents’ availability and convenience, considering the prevailing COVID-19 pandemic situation. The interviews were conducted in June–August 2021 by experienced senior members of the research team and lasted for around 45 min to an hour. The sessions were audio-recorded. Data collection stopped when saturation was reached, and no new information emerged.

Data were stored in a secure database in the BRAC James P Grant School of Public Health, BRAC University server with access restricted to team members only. Transcripts of interviews and notes (in case participants refused to be recorded) and recordings were stored in a secure online Box folder that was password protected. Physical/hard copies of data (qualitative transcripts) were locked away, and keys secured. A coding system replacing first names was developed so that no names are linked to interviews and corresponding translated transcripts. All transcripts were coded and stored.

### 2.5. Data Analysis

Audio recordings of all the IDIs were transcribed ad verbatim. As the IDIs were conducted in Bangla, the interviews were later translated and transcribed into English. Before coding, the data was read and reread to identify possible patterns and emerging themes. This was an iterative process applying the Anderson framework for analysis [20]. All interviews were coded inductively; a country-specific coding manual was developed for this purpose with the support of partners using the global codebook. The deductive codes were derived from literature, the global codebook, the interview guides and the research question. The deductive coding allowed for identifying and exploring country-specific new themes from the data and refining the predetermined codes. The study team members reviewed codes to ensure consistency of information. A data display matrix was developed for data extraction. Data were disaggregated into relevant themes and sub-themes and synthesised using information from the interviews and document review to ensure internal validity. Both approaches were used to find out the most common themes for necessary actions by the stakeholders in policy and practice.

## 3. Findings

### 3.1. Setting the Context

#### 3.1.1. Respondent Profile

We conducted 17 IDIs with stakeholders from various organisations knowledgeable and involved with AMR-related activities in Bangladesh. Out of the 17 stakeholders, ten were government officials, and seven were from various private organisations. Their age ranged from 30 to 67 years; most were male, and only four were females. Their highest level of education ranged from a master’s degree (MPH/MSc.) to a PhD. They had 5 to 40 years of experience in their profession (Median and IQR of work experience were 20 years and 9 years respectively) and had at least six months to 27 years of work experience in their current organisations (Median age was 5 years).

#### 3.1.2. Framing of AMR

According to the stakeholders, although a biological mechanism causes antimicrobial resistance, its consequence is eventually a social issue. The misuse and abuse of antibiotics is increasing as a consequence of modern health systems that rely largely on antibiotics to control infections. Antibiotic resistance develops in bacteria, not people or animals. These bacteria can infect both humans and animals, and these infections are more challenging to treat than those arising from non-resistant bacteria. Antibiotic resistance incurs medical expenses, lengthens hospital stays, and raises mortality. One main issue that came up repeatedly during the discussion was the lack of data to gauge the problem properly.

#### 3.1.3. AMR and COVID-19

The OTC sale of antimicrobials and its excessive use and consequent AMR has become a burning public health issue. The situation has worsened during the COVID-19 pandemic as the irrational use of antibiotics has increased. If this situation continues, there would be a significant impact on health; however, it also opens an opportunity to improve infectious disease control conditions in the country.


*“COVID-19 is a visible pandemic whereas AMR is a silent one…due to increased use of these drugs, the people of Bangladesh will suffer a lot as no medicine (antibiotics) is going to work for them anymore.”*
(R16)

### 3.2. Development and Implementation of NAP

The Director of Disease Control and Line Director, CDC/DGHS was selected as the focal agency to coordinate the national program. National-level committees were formed to ensure multisectoral involvement, including the inter-ministerial National Steering Committee (NSC) (chaired by the Minister) and the National Technical Committee (NTC) (chaired by the Director-General of Health Services). The NSC and the NTC devised and adopted a National Strategy for Antimicrobial Resistance Containment in Bangladesh and drafted a NAP in alignment with GAP for the 2017–2022 period [21]. This plan was again revised for 2021–2026 [22]. Although the NAP was a multisectoral approach and initiative, three key action plans stipulated in NAP were surveillance, (operation) research and optimisation of antimicrobial use (Figure 2).

To facilitate the development of the NAP, a multisectoral working group comprising personnel from human health, animal health and drug administrations was formed. The respondents mentioned that the three sectors of the human health, animal/livestock and fisheries were closely involved from the very beginning:


*“During the policy design of NAP, various sectors were involved. But only the three sectors which were actively working for NAP designing were the human health, animal (livestock), and fisheries sectors.”*
(R2)

The SMART (Specific, Measurable, Achievable, Relevant, and Time-Bound) indicators were mentioned in the roadmap of NAP to monitor progress; however, NAP-related activities were non-functional at the field level. The CDC/DGHS mostly had organised meetings and sectorial coordination. Due to COVID-19, most of the meetings were held virtually and irregularly: 


*“All the meetings are organised by CDC, DGHS since the design of NAP began, which used to hold on a regular basis—due to COVID-19 pandemic, no sectorial meetings were arranged regularly…only the virtual meetings held once/twice during the whole pandemic.”*
(R4)

#### 3.2.1. Implementation Tool

##### Awareness-Building

In Bangladesh, guidelines or protocols related to the containment of AMR, and/or the use of antibiotics were not included in the medical or other allied health curricula. Key informants mentioned that some institutions, e.g., BSMMU, and Chittagong Medical College, follow their institutional guidelines to treat patients. Some educational/awareness programmes such as training for the health care providers have been organised by the Food and Agriculture Organization (FAO). They have initiated training for physicians, livestock services providers and public health experts since 2016, following their developed guidelines on the containment of AMR. So far, they have provided training to 461 professionals (231 physicians and 230 veterinary doctors). Experts from the Food and Agriculture Organization (FAO) mentioned that this was the first training programme that brought together the human and animal sector professionals for awareness-building on antimicrobial use and resistance, including containment measures:


*“As per my knowledge, there is no national guidelines/protocol to be followed in the hospitals for AMR containment…few of the hospitals follow guidelines prepared by their respective facility experts.”*
(R3)

##### Awareness-Building among Hospital Patients and Attendants

The government initiated some hospital-based awareness programs targeting the patients and their attendants. The objective of these programmes was to distribute leaflets, arrange rallies and display posters, banners and announcements regarding the prevention of irrational use of antibiotics. However, experts mentioned that due to COVID-19, these did not happen in the last year:


*“Awareness on AMR containment usually held during the AMR weeks… as per my knowledge, no such programmes to aware the community people regarding the AMR issues.”*
(R11)

##### Infection Prevention and Control in Health Facilities

The US CDC offered various training for healthcare providers (HCP) in different hospitals, including physicians, nurses and support staff, including ToT (Training of Trainers) training in collaboration with the WHO [23]. The CDC has developed a guideline on Infection Prevention and Control (IPC) and is trying to provide training to endorse and activate the system, particularly in public hospitals. In addition, the USAID MTaPS (Medicines, Technologies and Pharmaceutical Services) Programme provided training for the hospital staff in their working areas in Comilla, Munshiganj, Nilfamari and Rangpur (total staff trained = 140). To quote:


*“IPC is an important component to reduce the AMR …some small scales on-going IPC programmes are running in the country, however, these need to be done on a large scale.”*
(R1)


*“I think, the workforce from livestock sectors also need to be trained on IPC… they also lack knowledge…due to lack of training on AMR-related issues.”*
(R2)

#### 3.2.2. Monitoring and Evaluation

Key respondents from different sectors informed that there was no national monitoring and evaluation framework for NAP activities as yet. However, the Director-General of Drug Administration (DGDA), with the support of the USAID MTaPS Program, was reported to develop such a framework. Some monitoring and evaluation of sectoral activities are currently running, such as prescription monitoring by DGDA and surveillance by CDC, DGHS through national reference labs.

In February 2020, the National technical committee of The Institute of Epidemiology, Disease Control and Research (IEDCR) was selected as the AMR Surveillance Coordination Center for human health and surveillance activities, started in nine sites all over Bangladesh in two phases. Surveillance physicians enrolled patients with five different types of infectious conditions according to case definitions and collected a specific number of defined samples from these patients. According to the GLASS (Global AMR Surveillance System) protocol of WHO, ten priority pathogens were selected considering the country context. To speed up the process, the head of each surveillance team was also the head of the microbiology section of that particular institution. The project facilitator and the microbiologist from IEDCR coordinated communication with the site teams and sent necessary feedback. The IEDCR team also visited the surveillance sites from time to time for field-level monitoring. According to respondents, the process was still at a preliminary stage and needed expansion with specified roles and responsibilities of the parties for transparency and accountability:

*We are trying to maintain the M&E indicators in our implementation. We have a team for this. Though we could not start the work indicator-wise, we are still trying to follow the indicators*.(R7)

The integrated M&E activities of the country were assessed as neither specific and measurable nor time-bound [24]. The DGHS was responsible for monitoring NAP activities but needed support from other stakeholders. Since Bangladesh is at an early implementation phase of NAP on AMR, experts believed that it was too early to comment on the programme’s effectiveness: 


*“To understand the effectiveness of the programmes, we need to wait for a few more years…”*
(R15)

Moreover, no mechanism for assessment on the effectiveness of antimicrobial stewardship programmes (ASPs) was available in Bangladesh:


*“Stewardship is part of NAP, but no one has any idea about it. Drug sellers don’t follow guidelines to prescribe medicines; our people are also not aware of the disadvantages of antibiotics use.”*
(R16)

A rapid review of the available literature found that NAP strongly aligned with the GAP (Rapid Review) [25]. In the veterinary sector, there were policy gaps related to an explicit financing modality, specifications for antimicrobial stewardship and rigorous operational and M&E framework [24].

#### 3.2.3. Feedback Mechanisms and Reporting

The experts suggested that feedback mechanisms and proper reporting systems on NAP activities are essential besides M&E, which were lacking in Bangladesh. AMR surveillance was operating in nine facilities, and the CDC, DGHS provided an annual report to the WHO based on surveillance system Data from IEDCR. The respondents opined that a regular feedback and reporting mechanism was direly needed: 


*Feedback mechanism and reporting are*
*very crucial… without these two, we can’t understand the status of surveillance or other activities.*
(R6)

From a rapid review of available literature, we found that AMR surveillance findings were disseminated only once to the stakeholders and policymakers at IEDCR in November 2019. The graphical representation of AMR surveillance data was updated in real-time from the surveillance sites. (http://119.148.17.100:8080/amr/summary_graph.php; accessed on 17 May 2022).

#### 3.2.4. Sustainability

Experts mentioned that the NAP should be revised with appropriate M&E mechanisms and adequate funding to ensure sustainability. Most of them opined that the AMR activities and NAP on AMR would not be sustainable without sufficient funds; for this, government funding was needed, and it should not depend on donors: 


*“Sustainability of the NAP is not ensured because it depends on funding.”*
(R2)

On the other hand, some respondents mentioned that many donor agencies were currently providing funds for AMR activities, but the budget was not used properly. Most public sectors did not have a dedicated budget for AMR activities, including the animal sector.

#### 3.2.5. One Health Engagement

According to the KIs, the One Health approach for integrating the health, animal and environment sectors to contain AMR is crucial. While the human, animal and fisheries sectors were working for it, the environment sector was yet to be involved in AMR-related activities. They suggested that one health engagement would be possible if the government took the initiative to engage all relevant sectors: 


*“One health approach is a must to achieve the goals of NAP… environment sector in Bangladesh is yet to take any initiative for containing AMR… so I would say, all sectors should come forward…”*
(R3)

### 3.3. Constraints in Implementing NAP

#### 3.3.1. Shortage of Health Workforce

One of the common challenges faced universally by all sectors was a shortage of the relevant workforce. The representative from CDC, DGHS informed that only three persons from DGHS were dedicated to implementing NAP:


*“Shortage of workforce is a common scenario for all sectors… need more people to engage in AMR-related work; otherwise, relevant tasks would not be accomplished on time…”*
(R11)


*“We have only three dedicated staff for AMR-related activities…we can’t achieve our goals with this small workforce.”*
(R1)

#### 3.3.2. High Turnover

Dedicated workforces have frequently been transferred from one department to another. Therefore, new staff used to join regularly and engaged in AMR-related activities afresh, with the additional problem that some might not even be interested in working in this area. Further, the personnel involved in designing the NAP were not engaged in the implementation phase, which made the tasks challenging due to a lack of hindsight:


*“Staff is switching (some are retired, some are being transferred to other departments, so new employees keep joining all the time… they may not even be interested in AMR containment activities. Some of them also take time to adapt with the NAP on AMR activities.”*
(R2)

#### 3.3.3. Engagement in COVID-19 Emergency Management

Due to the COVID-19 pandemic, AMR-related activities receded in priority. Most of the staff in the public sector hospitals were engaged in COVID-19 related activities, which was another constraint handled by several departments:


*“Most of our staff are involved with COVD-19 related tasks. We have staff shortage even in normal days… now the number of dedicated staff for AMR activities is too less like 2 or 3 persons are working dedicatedly, but it’s not enough.”*
(R1)

Besides this, they suggested that indiscriminate use of antibiotics for presumptive treatment of COVID-19 by clinicians and laypeople, especially at the beginning of the pandemic, might have worsened the AMR situation.

#### 3.3.4. Financial Constraints

Respondents from public sectors expressed that financial constraint was a major barrier to implementing NAP activities. According to them, dedicated government funds were needed to implement the NAP activities for the containment of AMR, and donor funds were not enough. Some respondents from the private sector also echoed them. However, they also opined that the funding available from different donor agencies needed proper utilisation:


*“We have financial constraints in almost all sectors… more budgets needed to implement the NAP activities … funds should not depend on donors, our government. should provide funds for AMR-related activities…”*
(R7)


*“The budget is minimal for AMR-related activities in many sectors… no dedicated fund for AMR containment in our department… we will submit a budget soon.”*
(R2)

### 3.4. Stakeholders Recommendations

The KIs recommended some actions based on their experiences with implementing the NAP:

#### 3.4.1. Strengthening Multi-Sectorial Coordination

According to KIs, strengthening multisectoral coordination is essential to improving NAP and AMR-related activities. All relevant sectors (human, animal, fisheries, environment and agriculture) must be involved in AMR-related activities, and coordination across the sectors was crucial:


*“We (all sectors) are not working together to implement the NAP on AMR…active participation/involvement from all sectors are important. Otherwise, the implementation plan would be useless.”*
(R7)

#### 3.4.2. Revision/Update of the NAP Strategy in the Context of COVID-19 Pandemic

The KIIs pointed out that proper implementation would not be possible without revision or updating of the recent version of NAP following the COVID-19 pandemic. 


*“The current version of NAP is old… We need to revise the version soon… the next version is under development although.”*
(R4)

#### 3.4.3. Proper Documentation for Tracking the Progress of NAP on AMR Activities

Surveillance data and other NAP activities should be documented appropriately and published regularly. Other than the findings from on-going surveillance data, progress in activities such as knowledge dissemination or advocacy was not visible. Based on evidence-based results, further decisions can improve the movement. Due to the absence of proper documentation of NAP-related activities in Bangladesh, policymakers were unable to gauge the progress of the work:


*“Even our policymakers don’t know about the NAP for containment of AMR-related work progress. Without proper documentation, none of them can understand the current situation, neither can decide the next steps.”*
(R6)

## 4. Discussion

AMR is a growing public health issue in Bangladesh, and the urgency to contain it cannot be overemphasised. It has been five years since the National Action Plan (NAP) on containing AMR in Bangladesh was put in place; however, no studies have been conducted yet regarding its implementation progress. This exploratory study based on in-depth interviews of the key stakeholders attempted to fill in this knowledge gap, understand the dynamics of NAP development and implementation and consolidate their experiences and learning to take this forward. The findings reveal the slow progress of the NAP activities, constrained by the shortage of a trained and motivated health workforce and financial resources, as well as the COVID-19 pandemic. These are discussed below with context to draw up some key conclusions.

AMR in Bangladesh is more of a social issue than a medical/health issue due to the easy availability of antimicrobials over the counter (OTC) and their irrational use [15,26]. Like other OTC medicines, their consumption is without any regulation e.g., by prescription or ethics [17]. The poor awareness of the problem compounded this regulatory problem by healthcare providers of all sorts and their non-compliance with the containment of AMR protocols [16]. Besides this, the current pandemic led to the indiscriminate use of antimicrobials for questionable treatment of COVID-19 and added to the burden [27,28]. On the flip side, it also created new opportunities to control infectious diseases and AMR threats by meticulously implementing the NAP in Bangladesh. Clinicians need to be cognisant of the enhanced problem and motivated to strictly follow rational use principles/guidelines when using antibiotics for the treatment of COVID-19. The general public also needs to be educated on the futility of using antibiotics in treating COVID-19. 

Following GAP, the NAP was designed to integrate all sectors of the human and animal health and environment into a single One Health approach for surveillance, operations research and optimisation of antimicrobial use. This One Health approach is significant for properly implementing NAP in Bangladesh [16]. However, it is at a rudimentary stage, as observed in the veterinary AMR situation [24], and there are shortcomings in displaying the evidence-based stewardship role in articulating the COVID-19 coordinated response. Interestingly, the environmental sector was conspicuously absent in the ‘NAP-on-AMR’ related activities. Policymakers and practitioners have to devise ways to integrate these into the One Health platform to work together. Though some monitoring and evaluation activities and surveillance had been going on, those were primarily small in scale, uncoordinated, and lacking in a feedback mechanism.

The shortage of a dedicated and trained health workforce and financial resources for NAP implementation were two recurring themes emerging from the in-depth interviews. Equitable deployment of adequately trained/skilled human resources for health (HRH) is imperative for NAP implementation, even more so in a pandemic. The recommendations of different stakeholders from various fields should also be taken into account while implementing NAP, as it takes time to implement this kind of multisectoral action plan, especially in low-and middle-income countries (LMICs) such as Bangladesh. Besides this, the indiscriminate use of antimicrobials (and anti-parasital medicines) for unsubstantiated prevention and treatment of COVID-19 weakened the progress regarding containment of AMR in the country. 

Other basic issues such as customised awareness-building for both supply and demand sides and water, sanitation and hygiene measures for infection prevention and control warrant urgent attention from the policymakers and practitioners. Last but not least, political commitment, power dynamics and an understanding of local contexts are essential for any NAP on AMR to be successfully operationalised [29,30].

### Strengths and Weaknesses

This study is the first of its kind to assess the current status of the NAP on AMR implementation in Bangladesh. Findings from the study are expected to help policymakers and practitioners revise the NAP in the context of the current COVID-19 pandemic situation and expedite its implementation to contain the silent epidemic of AMR. The analysis was limited to the availability of relevant documents in the public domain. In addition, the pandemic situation limited our efforts to interview a broader and larger audience of stakeholders, which might have yielded more useful information. For example, we could not cover the clinicians in the hospital settings as they were too busy to give us an appointment for IDI. However, given that we largely succeeded in reviewing the main documents and interviewing the key players in both public and non-state sectors, including academia and research, we are confident that it adequately represented the real-life situation.

## 5. Conclusions

Five years passed since the development of NAP in Bangladesh, yet its implementation is not up to the level that the situation’s urgency requires. Key stakeholders in policy and practice need to understand the gravity of the problem, especially following the large-scale misuse of antibiotics during the COVID-19 pandemic, and should rise to the occasion with adequate human and financial resources before we find ourselves in the pre-antibiotics era. The information related to the AMR crisis also needs to be transparent and accountable for the people to get organised and demand urgent action [31].

## Figures and Tables

**Figure 1 antibiotics-11-00690-f001:**
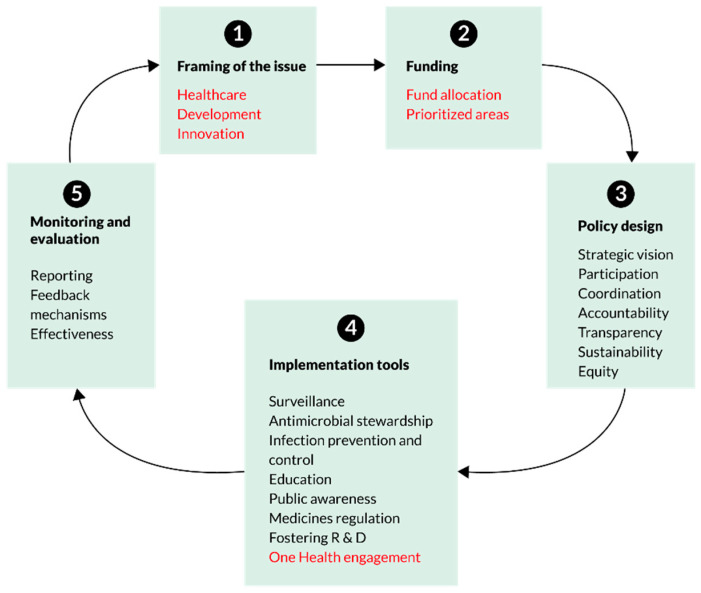
An adapted conceptual framework for assessment of National Action Plans [20].

**Figure 2 antibiotics-11-00690-f002:**
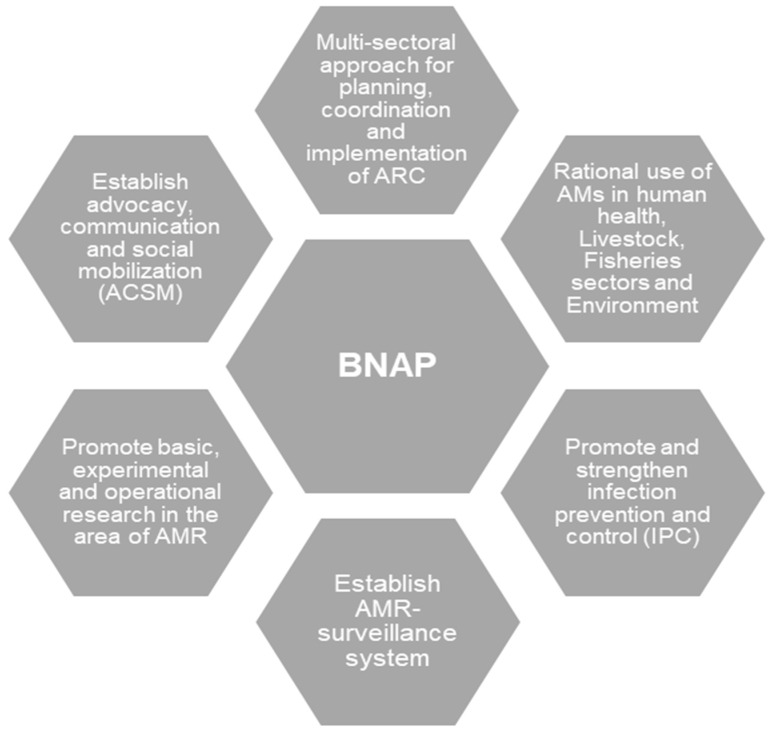
Bangladesh National Action Plan (BNAP): Antimicrobial Resistance Containment in Bangladesh 2017–2022 [21].

**Table 1 antibiotics-11-00690-t001:** Domains explored in the IDI guideline [14,20].

Framing of AMR Policy design and implementation of AMR Participation, coordination, transparency and accountability Sustainability and funding: Education and public awareness MR Surveillance system, Risk assessment, Monitoring, and Evaluation: Research and innovation Prevention and control of infection and One Health Management Antimicrobial stewardship programme, medicine regulation for national use of antimicrobials Impact of COVID-19 on AMR Stewardship Stakeholder analysis AMR related work experience

## Data Availability

The study dataset is available from the corresponding author upon reasonable request.

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
