# Peer review of "The Implementation of National Action Plan (NAP) on Antimicrobial Resistance (AMR) in Bangladesh: Challenges and Lessons Learned from a Cross-Sectional Qualitative Study"

_antibiotics, 2022, doi:10.3390/antibiotics11050690_

Round 1

Reviewer 1 Report

The manuscript is very interesting because it contextualizes the difficulties in conducting a complex program such as that of integrated interventions for the implementation of a national plan to combat antimicrobial resistance in a very effective way.
With great intellectual honesty and good competence the main obstacles to the full realization of programs of this type are then listed and analyzed. There is no lack of clear references to the need for strong commitment.

No other comments are needed

Author Response

Thanks to the reviewer.

Reviewer 2 Report

This is a well-written qualitative study describing the state of implementation of the NAP on AMR in Bangladesh since initial declaration 5 years prior to the study. By interviewing stakeholders, authors identify multiple challenges that need to be remedied for success of NAP on AMR in Bangladesh. I feel this study has a lot of value to readership, which is hopefully includes high-level officials within the Government, and policy makers from surrounding nations. I have a few suggestions:

  1. What is the authors' relationship to this work? Are they among the stakeholders or are they an independent group? Was this work commissioned by the government or an independent group? Some more background would help
  2. Abstract: the opening sentence needs to be made more succinct and clear. The goal/aims of the study are not currently clear as it is written but are clarified below in the text. 
  3. Introduction - would delete "of variable quality and efficacy" in the second sentence of the second paragraph
  4. Introduction - would remove the part about "food preservatives" in the second paragraph as I'm not sure the link to AMR is established.
  5. Figure 1: considering numbering the boxes so readers know where to start and finish
  6. 2.3 Data Collection and Management - 17 IDIs were conducted but what was the "denominator?" how many stakeholder groups were eligible to be interviewed? In survey studies, a denominator and a percentage completion should be included. Is 17 a lot or very few? This has implications for validity.
  7. Were any stakeholders hospital-based? it is not clear from the methods. It would be important to interview stakeholders doing the day-to-day work of patient care
  8. 3.3.2 - High Turnover. Please modify sentence ending in "dedicatedly" which is confusing in structure. 
  9. 3.3.3 - was there any connection made either in the qualitative questioning or by the respondents themselves about the link between COVID and AMR? This is a topic of great interest and therefore anything you can include would be beneficial either in this section or the prior section 3.1.3. You spend much of the discussion speaking about COVID and AMR but there is not much about it in the results. 
  10. Discussion - please expand on how the COVID pandemic has put additional emphasis on the need to create a successful NAP on AMR in Bangladesh - what aspects are even more critical now, and how should future revisions of the NAP account for what happened with AMR during COVID-19?

Author Response

This is a well-written qualitative study describing the state of implementation of the NAP on AMR in Bangladesh since initial declaration 5 years prior to the study. By interviewing stakeholders, authors identify multiple challenges that need to be remedied for success of NAP on AMR in Bangladesh. I feel this study has a lot of value to readership, which is hopefully includes high-level officials within the Government, and policy makers from surrounding nations. I have a few suggestions:

  1. What is the authors' relationship to this work? Are they among the stakeholders or are they an independent group? Was this work commissioned by the government or an independent group? Some more background would help

Response: An independent group of researchers from BRAC JPG School of Public Health conducted this study. This project was not commissioned by the government.

  1. Abstract: the opening sentence needs to be made more succinct and clear. The goal/aims of the study are not currently clear as it is written but are clarified below in the text. 

Response: The objective of the paper is now clearly written in the abstract (“This study explored the current situation of the National Action Plan (NAP) on Antimicrobial Resistance (AMR) implementation in Bangladesh and examined how different sectors (human, animal, and environment) addressed the AMR problem in policy and practice, associated challenges, and barriers to identifying policy lessons and practices”).

  1. Introduction - would delete "of variable quality and efficacy" in the second sentence of the second paragraph

Response: We have deleted “of variable quality and efficacy” as suggested.

  1. Introduction - would remove the part about "food preservatives" in the second paragraph as I'm not sure the link to AMR is established.

Response: Removed as suggested.

  1. Figure 1: considering numbering the boxes so readers know where to start and finish

Response: We have revised the Figure 1 as suggested.

  1. 2.3 Data Collection and Management - 17 IDIs were conducted but what was the "denominator?" how many stakeholder groups were eligible to be interviewed? In survey studies, a denominator and a percentage completion should be included. Is 17 a lot or very few? This has implications for validity.

Response: This study applied the qualitative research methods and conducted key informant interviews (KIIs); therefore, no denominator was needed. The stakeholders were eligible because they were experienced in AMR-related activities. Moreover, 17 interviews were enough because when we achieved the data saturation, no more interviews were needed to conduct the study.

  1. Were any stakeholders hospital-based? it is not clear from the methods. It would be important to interview stakeholders doing the day-to-day work of patient care

Response: Stakeholders were from different research institutes, donor agencies, and academicians of different medical college hospitals who were involved with AMR -related research activities. Two stakeholders were from two different Medical College Hospitals, but they were not involved in hospital in-patient care. They were from the department of Pharmacology and Department of Microbiology.

  1. 3.3.2 - High Turnover. Please modify sentence ending in "dedicatedly" which is confusing in structure. 

Response: The term “dedicatedly” is removed from the sentence as suggested.

  1. 3.3.3 - was there any connection made either in the qualitative questioning or by the respondents themselves about the link between COVID and AMR? This is a topic of great interest and therefore anything you can include would be beneficial either in this section or the prior section 3.1.3. You spend much of the discussion speaking about COVID and AMR but there is not much about it in the results. 

Response: We agree with the reviewer…added a sentence to this!

Discussion - please expand on how the COVID pandemic has put additional emphasis on the need to create a successful NAP on AMR in Bangladesh - what aspects are even more critical now, and how should future revisions of the NAP account for what happened with AMR during COVID-19?

Response: The connection between the AMR situation and COVID-19 i.e., worsening of AMR situation due to indiscriminate use of antibiotics during the pandemic is already discussed in the last part of the 2nd para of the Discussion section (“Besides, the current pandemic led to indiscriminate use of antimicrobials for questionable treatment of COVID-19 and added to the burden [27,28]”. On the flip side, it also created new opportunities to control infectious diseases and AMR threats by meticulously implementing the NAP in Bangladesh. The clinicians need to be cognizant of the enhanced and motivated to strictly follow rational use principles/guidelines when using antibiotics for treatment of COVID-19. The general public also need to be educated on the futility of using antibiotics in treating COVID-19.”

Reviewer 3 Report

I believe this is an important study on the matter of implementation of a national plan for antimicrobial resistance. This is an emerging issue worldwide and articles like this raise awareness in different parts of the world. I am happy that references are appropriate and up to date and that tables and figures are well presented.

my suggestions are as follows:

  1. add study design to the title
  2. add full names of NAP, AMR, WHO, and ME in abstract
  3. add study period in abstract
  4. Southeast Asia is considered to have the highest risk of AMR from the introduction - this is an interesting statement, please explain it further
  5. add ethical approvement to section 2.3.
  6. add median and IQR on years of experience in 3.1.1.
  7. rephrase for clarity first sentence in 3.1.2.
  8. 3.1.3. OTC already introduced as abbreviation
  9. 3.2. NAP already introduced
  10. please add other possible biases to 4.1.

Author Response

I believe this is an important study on the matter of implementation of a national plan for antimicrobial resistance. This is an emerging issue worldwide and articles like this raise awareness in different parts of the world. I am happy that references are appropriate and up to date and that tables and figures are well presented.

my suggestions are as follows:

  1. add study design to the title

Response: The study design is included in the title. The revised title is “The Implementation of National Action Plan (NAP) on Anti-microbial Resistance (AMR) in Bangladesh: Challenges and Lessons Learned from a cross-sectional qualitative study”.

  1. add full names of NAP, AMR, WHO, and ME in abstract

Response: We have included the full abbreviations of NAP, AMR, WHO, and ME in the abstract and highlighted them in yellow. m

NAP= National Action Plan

AMR= Antimicrobial Resistance (AMR)

WHO=World Health Organization

M&E=national Monitoring and Evaluation

  1. add study period in abstract

Response: The study period is mentioned in the abstract. This study was conducted from January 2021 to December 2021.

  1. Southeast Asia is considered to have the highest risk of AMR from the introduction - this is an interesting statement, please explain it further

Response: We think that this is adequately covered in the 3rd paragraph of introduction and suffices for this paper. We are afraid of boring the readers with duplication, repetition and unnecessary expansion here.

  1. add ethical approvement to section 2.3.

Response: Ethical approval is added to section 2.3.

  1. add median and IQR on years of experience in 3.1.1.

Response: Median and IQR on the years of experience are included in section 3.1.1 and these were 20 years and 9 years respectively.

  1. rephrase for clarity first sentence in 3.1.2.

Response: The sentence is rephrased as suggested.

  1. 3.1.3. OTC already introduced as abbreviation

Response: We have made the necessary changes.

  1. 3.2. NAP already introduced

Response: We have made the correction as suggested.

  1. please add other possible biases to 4.1.

Response: Added in 4.1>(“In this regard, we could not cover the clinicians in the hospital settings as they were too busy to give us appointment for IDIs”.)

Round 2

Reviewer 2 Report

Opening sentence is more clear now but still a bit rambling and could be made into 2 sentences. Otherwise no major issues with revised version